# CAN I UNDERSTAND WHAT I CREATE? SELF-KNOWLEDGE EVALUATION OF LARGE LANGUAGE MODELS

## ABSTRACT

Large language models (LLMs) have achieved remarkable progress in linguistic tasks, necessitating robust evaluation frameworks to understand their capabilities and limitations. Inspired by Feynman's principle of understanding through creation, we introduce a self-knowledge evaluation framework that is easy to implement, evaluating models on their ability to comprehend and respond to self-generated questions. Our findings, based on testing multiple models across diverse tasks, reveal significant gaps in the model's self-knowledge ability. Further analysis indicates these gaps may be due to misalignment with human attention mechanisms. Additionally, fine-tuning on self-generated math task may enhance the model's math performance, highlighting the potential of the framework for efficient and insightful model evaluation and may also contribute to the improvement of LLMs.

## 1 INTRODUCTION

In recent years, large language models (LLMs) have reached groundbreaking milestones, significantly advancing in areas such as semantic understanding, sentence translation, and more OpenAI (2023); Touvron et al. (2023); Anil et al. (2023); Team et al. (2023). These models not only facilitate enhanced interaction between computers and human language but also drive innovation across numerous applications. However, as these models become increasingly central to technological advancements and their applications more widespread, it is crucial to establish robust, systematic evaluation frameworks. Such frameworks are essential not only for understanding the full spectrum of capabilities these models possess but also for identifying their limitations and potential biases.

The evaluation of large language models has made significant strides in recent years, with researchers developing numerous benchmarks aimed at testing various aspects of model performance (Hendrycks et al., 2020; Li et al., 2023; Zhong et al., 2023). However, the current evaluation methods still have notable shortcomings. Firstly, most benchmarks require substantial human and material resources and often necessitate the involvement of domain experts to accurately assess correctness. Secondly, evaluations that measure a large model's capability through self-evaluation of its own knowledge is less explored. This gap highlights the need for developing more efficient and insightful evaluation techniques that not only reduce the dependency on extensive resources but also enhance the models' ability to evaluate their own performance and limitations.

Motivated by Richard Feynman's famous quote: "What I cannot create, I do not understand." We would like to evaluate the large language model's capability through its "reverse version", i.e. does the model really understand the questions and solutions created by itself?, which we termed the self-knowledge of the model. This capability is effectively realized by a *truthful* human, since the originator of a question and its corresponding answer should be able to respond consistently and without difficulty if asked the same question by others if they truly comprehend this knowledge. This ease comes naturally from being the initial creator of the question, so when evaluated on a benchmark generated in this way, a self-knowledgable model should receive an accuracy of nearly $100\%$ easily.

In this paper, we provide a novel framework that can evaluate the model's self-knowledge ability and is very **easy to implement.** We conduct an extensive evaluation of 7 popular LLMs across 9 tasks, including counting words, math, theorem proving, etc. We also conduct evaluation on large multi-modal models (LMMs). We summarize some of our findings as follows:

- We find that modern LLMs and LMMs have unsatisfactory behaviors on self-knowledge evaluations, which is far from perfect.
- By analyzing a designated word counting task, we find that models become much similar to the human-inspired attention-based mechanisms when the model gets a higher self-knowledge score. The poor self-knowledge task performance may be explained by *additive effect* of misalignment with this attention-based mechanism and the less-concentrates of LLM attention than humans.
- We find only GPT-4 and Gemma achieve $100\%$ accuracy when the question-generating process is given in context and their accuracy is reduced when the context is added with noisy contents. GPT-4 has accuracy less reduced than Gemma, making GPT-4 has more similar behaviour like humans than other models.
- We find that fine-tuning the data generated by the self-knowledge math task may improve the performance on GSM-8k.
- We find that expert-based prompts may usually improve self-knowledge ability but chain-of-thought prompting may usually not.

## 2 RELATED WORKS

**Evaluation of large generative models.** Recent years have seen significant advancements in the development of large generative models, including large vision models (LVMs) Radford et al. (2021); Kirillov et al. (2023), large language models (LLMs) OpenAI (2023); Touvron et al. (2023); Bai et al. (2023); Team et al. (2024); Jiang et al. (2023a), and their evolution into large multi-modal models (LMMs) OpenAI (2023); Liu et al. (2023); Zhu et al. (2023); Wei et al. (2023); Koh et al. (2023); Ge et al. (2023), demonstrating near-human proficiency and even a spark of AGI. Evaluation of these large generative models is a fast-evolving field across various tasks, datasets, and benchmarks Zhong et al. (2023); Yue et al. (2023); Fu et al. (2023); Wei et al. (2024); Sun et al. (2024). It encompasses a wide range of domains, including the generation of language, images, videos, and audio. However, there is a lack of evaluations that measure a large generative model's self-knowledge of its own capabilities. Specifically, we focus on the self-knowledge evaluation of LLMs that can understand instruction and output responses, as well as LMMs that can both understand images and generate images.

**Evaluation of LLM's instruction-following ability.** Several studies have established benchmarks for evaluating LLMs' instruction-following abilities. Jiang et al. (2023b) proposed FollowBench that sequentially add fine-grained constraints to construct multi-level instructions. Zhou et al. (2023) emphasized objective evaluations with verifiable instructions. Meanwhile, Qin et al. (2024) constructed a benchmark composed of several distinct instructions and decomposed questions for the assessment of the instruction following. These benchmarks require manually constructing a large number of instructions and answers. Differently, our work mainly focuses on the large model's self-knowledge of its own capabilities, which is also independent of collecting additional annotated answers.

## 3 THE SELF-KNOWLEDGE EVALUATION FRAMEWORK

To evaluate the self-knowledge of large language models (LLMs), we propose a novel method called **First Generate, Then Evaluate**, inspired by the concept of "self-questioning and answering." This approach involves a two-step process:

1. The self-generate step: We utilize a question-generating prompt to instruct the LLM to produce relevant content. The model provides answers that are either specified by the prompt or generated autonomously.

$$\text{LLM(question-generating prompt)} \xrightarrow{\text{generate}} \mathbf{x}; \mathbf{a}, \tag{1}$$

where $\mathbf{x}$ is the generated paragraph and $\mathbf{a}$ is the corresponding answer specified by the prompt or generated by the model directly.

2. The self-verify step: It uses a question-verifying prompt to assess the previously generated content $\mathbf{x}$:

$$\text{LLM(question-verifying prompt}, \mathbf{x}) \xrightarrow{\text{generate}} \hat{\mathbf{a}}, \tag{2}$$

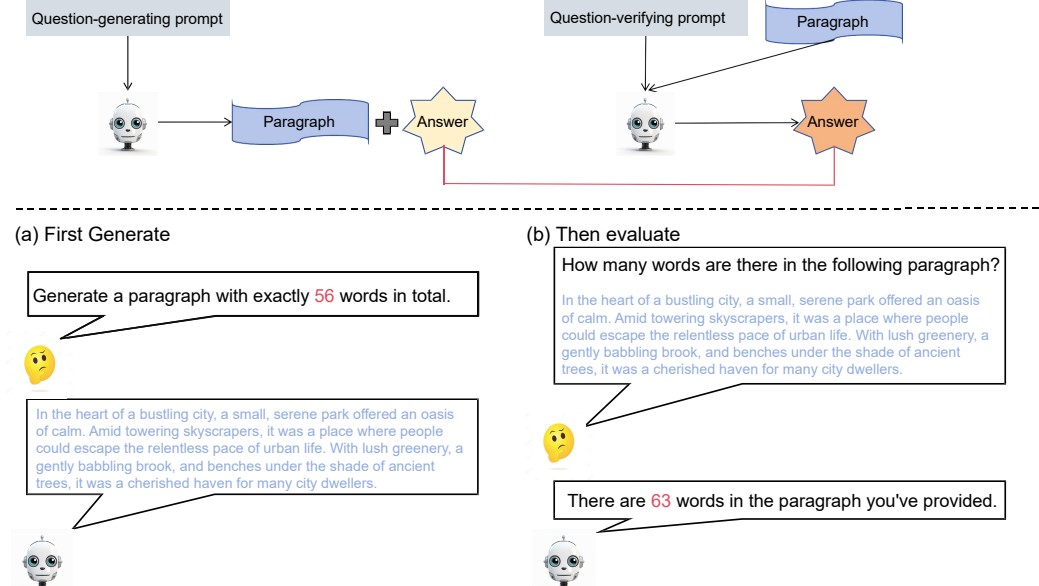

Figure 1: A case of "first generate, then evaluate". The model is first asked to generate a paragraph with 56 words. Then we can ask the model in a separate run and ask how many words are there in the previously generated paragraph. If the answer is not 56, we will raise an error.

where $\hat{\mathbf{a}}$ is the answer of question $\mathbf{x}$ under the verifying prompt. Note the question-generating prompt and the question-verifying prompt are pairing prompts that are designed to correlate with some ability of the model, and thus can be seen as evaluating the model's self-knowledge on a specific task. Then, the self-knowledge score is calculated by $\mathbb{I}(\mathbf{a} = \hat{\mathbf{a}})$. For $n$ pair of question-generating and question-verifying prompts, denote their respective answers be $\mathbf{a}_i$ and $\hat{\mathbf{a}}_i$, the self-knowledge score is calculated by $\frac{1}{n}\sum_{i=1}^{n}\mathbb{I}(\mathbf{a}_i = \hat{\mathbf{a}}_i)$. In this paper, we mainly consider the simplest self-evaluation strategy by directly asking the model to respond with the answer, more sophisticated self-verifying strategies like (Weng et al., 2022) are left for future work.

We have also presented a schematic view of the pipeline of our method in Figure 1. The question-generating prompt is depicted in Figure 1(a)'s self-generate process as "Generate a paragraph with exactly 56 words in total.". As LLM has strong instruction-following and writing abilities, it will generate a paragraph $\mathbf{x}$. Note the answer $\mathbf{a}$ for this word counting task is already contained in the prompt, i.e. $\mathbf{a} = 56$. Then Figure 1(b) shows the self-verify step, the question-verifying prompt is "How many words are there in the following paragraph?" and the model generates an answer of $\hat{\mathbf{a}} = 63$. The inconsistency of the answers $\mathbf{a} = 56$ and $\hat{\mathbf{a}} = 63$ gives rise to a case of not comprehending the self-knowledge. For more experiments in this manner, please see section 4.2.

One might wonder whether it's necessary to generate new samples every time we assess self-knowledge in a task. In other words, can we reuse previously generated samples for new tasks? The technical part here is that we can only access the generated paragraph $\mathbf{x}$ but do not have the task-specific answer $\mathbf{a}$. Fortunately, one can evaluate this case using the idea of **consistency.** Suppose $\mathbf{x}$ is generated by a question-generating prompt corresponds to task $T'$ and we want to evaluate the self-knowledge on task $T$ ($T' \neq T$). Suppose a transformation $\tau$ makes the answer to task $T$ **unchanged** when applying $\tau$ to $\mathbf{x}$, then the self-knowledge score can be calculated via

$$\mathbb{I}(\text{LLM}(\text{question-verifying prompt}, \mathbf{x}) = \text{LLM}(\text{question-verifying prompt}, \tau(\mathbf{x}))). \quad (3)$$

We have also presented a schematic view of a preposition counting task in Figure 2. Given a sample $\mathbf{x}$, we consider a question-verifying prompt as "How many prepositions appear in the following paragraph?". The answer to this question with respect to sample $\mathbf{x}$ is 14. Note an easy transformation $\tau$ to $\mathbf{x}$ will preserve the total number of prepositions in the paragraph, i.e. move the first sentence of the paragraph to the end of the paragraph. The inconsistency of the answers $\mathbf{a} = 56$ and $\hat{\mathbf{a}} = 63$ gives rise to a case of not comprehending the self-knowledge. For a dataset consisting of $n$ samples,

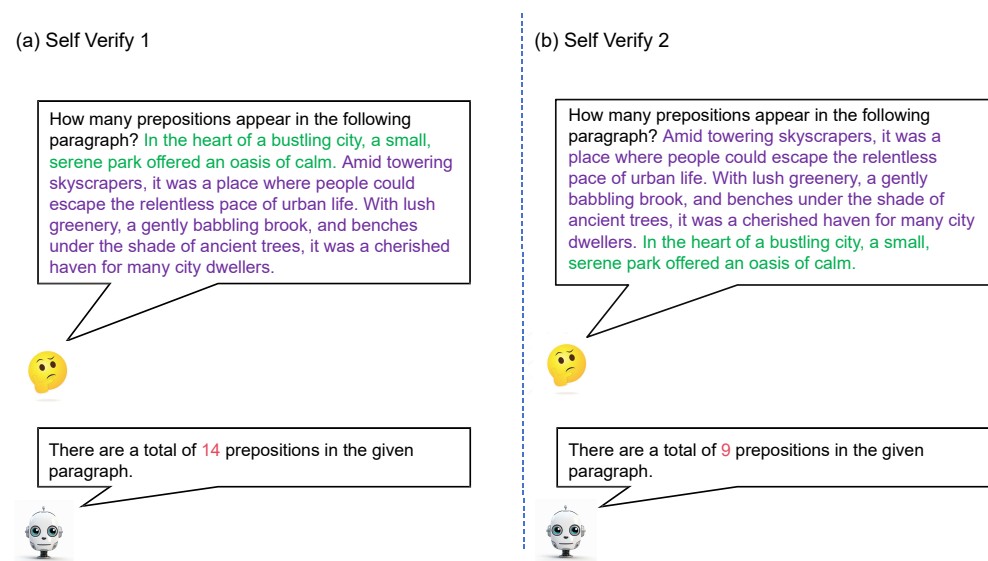

Figure 2: A case of using existing generated content. The model is first asked about the number of prepositions in its previously generated content. Then we cut the first sentence in the previous paragraph and paste it at the last and generate a new paragraph. Then we ask the model in a separate run about the number of prepositions in the newly generated paragraph. If the answer is not consistent, we will raise an error.

the self-knowledge score is the average of each sample's score. For the experiments in this spirit, please see section 4.4.

# 4 EVALUATING THE SELF-KNOWLEDGE OF LLMS

## 4.1 IMPLEMENTATION DETAILS

In our evaluation of language models, we incorporate seven widely recognized LLMs, each distinguished by its unique characteristics and training methodologies: GPT-3.5 (gpt-3.5-turbo-1106), GPT-4 OpenAI (2023) (gpt-4-0125-preview), Llama3-8B-Instruct, Llama2-7B-Chat Touvron et al. (2023), Mistral-7B-Instruct-v0.2 Jiang et al. (2023a), Gemma-1.1-7B-Instruct Team et al. (2024) and Qwen1.5-7B-Chat Bai et al. (2023). For API-based models (GPT-3.5 and GPT-4), we set the temperature to zero for stable generation. For open-sourced models, we follow their default generation strategy. We present all the evaluation results in Table 1, the detailed evaluation strategy will be discussed in the following subsections and the template questions can be found in Table 8 in the Appendix.

## 4.2 FIRST GENERATE, THEN EVALUATE

In this case, we mainly consider the answer to the generated question is designed to be known in advance, we use this way because asking the model to generate *both* the question and answers may limit the diversity of answers and sometimes even generate duplicate contents.

### 4.2.1 COUTING THE TOTAL NUMBER OF WORDS

Currently, the most advanced large language models (LLMs) employ an autoregressive framework, generating each subsequent token one at a time. Although the tokens used by various tokenizers do not necessarily correspond directly to English vocabulary, the principle of sequentially counting each token should be relatively simple for LLMs, given their inherent design to process information token-by-token. Given this, one would assume that tasks such as total word counting would be

Table 1: The accuracies of different LLMs under various self-knowledge tasks.

| Model | Total count | Designate count | Fact | ArXiv | Math | Theorem | Code | Avg |
|---|---|---|---|---|---|---|---|---|
| GPT-4 OpenAI (2023) | 0.03 | 0.46 | 0.71 | 0.13 | 0.24 | 0.51 | 0.08 | 0.31 |
| GPT-3.5 | 0.00 | 0.16 | 0.68 | 0.09 | 0.58 | 0.49 | 0.51 | 0.36 |
| Llama3-8B-Instruct | 0.00 | 0.39 | 0.30 | 0.00 | 0.14 | 0.29 | 0.68 | 0.26 |
| Llama2-7B-Chat Touvron et al. (2023) | 0.00 | 0.34 | 0.65 | 0.00 | 0.88 | 0.83 | 0.16 | 0.47 |
| Mistral-7B-Instruct-v0.2 Jiang et al. (2023a) | 0.00 | 0.13 | 0.92 | 0.00 | 0.23 | 0.58 | 0.07 | 0.32 |
| Gemma-1.1-7B-Instruct Team et al. (2024) | 0.00 | 0.24 | 0.15 | 0.01 | 0.93 | 0.71 | 0.42 | 0.33 |
| Qwen1.5-7B-Chat Bai et al. (2023) | 0.01 | 0.10 | 0.77 | 0.01 | 0.57 | 0.58 | 0.84 | 0.41 |

straightforward for these models. We ask the model to generate paragraphs from a length of 50 to 149 and get 100 samples. When we conducted tests to evaluate their capabilities in this regard, we were surprised to discover that their performance was very poor. A pictorial view can also be found in Figure 1.

### 4.2.2 GENERATE PARAGRAPH THAT CONTAINS A SPECIFIC NUMBER OF DESIGNATED WORDS

Theoretically, the task of generating a paragraph that contains a specific number of *designated* words should be well within the capabilities of autoregressive large language models (LLMs). Given that these models generate text sequentially, they inherently have the ability to review their own history, including tracking the frequency of specified terms as they generate new content. This capability should enable them to adjust their output to meet predefined criteria, such as incorporating a certain number of specific words. We ask the model to generate a designated "keyword" a predefined number of times and get 100 samples. Then in a separate run, we ask the model the appearance time of this specific keyword and check whether it is the same with our predefined frequency. We only consider the simplest case of only one keyword and leave the combination of multiple keywords as future work. The selection of keywords is flexible, one may randomly pick it from a dictionary or ask an LLM to pick from a summarization of new web content. However, despite these theoretical capabilities, our empirical tests reveal that the performance of these models remains unsatisfactory in executing this seemingly straightforward task. This underperformance suggests potential limitations in their current training or architectural design, which may not fully support dynamic adjustments based on historical data analysis during text generation.

### 4.2.3 FACTS

Testing models on their ability to accurately recall important dates related to historical figures is crucial because it assesses their precision in handling factual information. Remembering key dates, such as births, deaths, and significant events linked to these individuals, is essential for a reliable understanding of history. This precision is not just about storing data but also about the ability to retrieve it accurately when needed. Such tests are particularly important in educational contexts, where precise historical facts are fundamental for teaching and learning. They help ensure that AI models can serve as dependable resources for students and researchers who rely on accurate historical data. We ask the model to name a celebrity that was born on specific dates. Then in a separate run, we ask the model if the celebrity was born on this day. We generate 100 different days and the evaluation result shows that models usually show good consistency under this test.

### 4.2.4 ARXIV

ArXiv *dataset* is part of the standard pertaining dataset Pile (Gao et al., 2020) and captures the technical knowledge in many scientific areas. Testing large models on their ability to accurately retrieve arXiv IDs is important because it assesses their precision and efficiency in handling specific, detailed queries within academic and scientific contexts. Such testing not only ensures that models can effectively navigate and extract precise information from vast databases but also highlights their utility in supporting scholarly work and literature review processes, where accuracy is paramount. We ask the model to generate the title and IDs of an arXiv paper in a specific month. Then in a separate run, we ask the model the arXiv ID of the previously generated paper title and check whether it is consistent with the previously generated ID. We generate 100 different months and the evaluation result shows that models perform poorly on this task.

### 4.2.5 MATH

Testing large models on their ability to solve math problems is crucial for evaluating their performance because these tasks require a combination of several complex cognitive skills Azerbayev et al. (2023b); Yu et al. (2023). First, the model must accurately understand the natural language and symbolic notations used in the problem, recognizing key information and its context. Then, it needs to translate all linguistic descriptions into a mathematical format, applying the correct operations and formulas. Finally, the model must manage and manipulate numerical data to reach a solution. This process tests not only the model's linguistic comprehension but also its logical reasoning and numerical accuracy, providing a comprehensive assessment of its capabilities across different domains of intelligence. We ask the model to generate a math question with typical encountered math question answers like 10cm or $\pi$ etc. and we generate 100 different samples. Then in a separate run, we ask the model whether the predefined answer is consistent with the previously generated question.

### 4.2.6 THEOREM PROVING

Evaluating large models on their ability to solve mathematical proofs is essential because it assesses more than just their mathematical knowledge—it evaluates their logical thinking and problem-solving skills Azerbayev et al. (2023a); Yang et al. (2024). Mathematical proofs require understanding complex concepts and linking them together through a series of logical steps. This type of testing checks if the model can not only follow these steps but also organize and articulate them clearly and effectively. By doing so, we can determine how well the model can handle complex, abstract ideas and if it can apply its knowledge to develop coherent, logical solutions. This insight is crucial for understanding the depth and breadth of the model's cognitive abilities, making it a comprehensive test of its overall intellectual performance. However, verifying the correctness of a proof may be too challenging. We find that inequalities are a good testbed for this task as many of them can be verified by computers automatically. We also consider the simplest case of single variables inequalities as inequalities involving multiple variables are hard to verify their correctness even by humans and we also generate 100 different samples. Then in a separate run, we ask the model whether the previously generated inequality is correct or not.

### 4.2.7 CODE

Testing large models on their ability to write code is crucial for understanding how well they can apply computer science concepts in real situations Roziere et al. (2023); Guo et al. (2024). This type of testing goes beyond just knowing programming language rules. It looks at whether the model can effectively break down problems, think through solutions logically, and turn those ideas into working code. This helps us evaluate how well the model can handle practical tasks in computing, showing its potential to work as an effective tool in technology and software development. Such tests are key for seeing how theoretical knowledge translates into actual, usable applications. In the experiments, we ask the model to generate a program that has its execution result given, for eg. 10 and we also generate 100 different samples. Then, in a separate run, we ask the model the executed result of its generated program and check whether it is consistent.

## 4.3 VERIFY USING DUAL-GENERATING STRATEGY

We can also make further verify by using the generated content $\mathbf{x}$ without direct access to the generated answer $\mathbf{a}$. We will use the following dual-generating strategy, by designing a dual prompt that make the model generate a new content based on the existing content $\mathbf{x}$ and if the generation is correct will have the same answer under the question-verifying prompt.

The schematic process works as follows:

$$\text{LLM(dual-generating prompt, } \mathbf{x}) \xrightarrow{\text{generate}} \mathbf{x}'. \tag{4}$$

$$\text{LLM(question-verifying prompt, } \mathbf{x}') \xrightarrow{\text{generate}} \hat{\mathbf{a}}'. \tag{5}$$

The self-knowledge score is calculated by $\mathbb{I}(\hat{\mathbf{a}} = \hat{\mathbf{a}}')$, where $\hat{\mathbf{a}}$ is given by equation (2).

For example, for the total word count task, a possible dual-generating prompt will be "Generate a paragraph with the same number of words with the following paragraph." and the grammar task is to

Table 2: Self-knowledge score using dual-generating strategy.

| Model | Total count | Designate count | Fact | Grammar | Math | Code | Avg |
|---|---|---|---|---|---|---|---|
| GPT-4 OpenAI (2023) | 0.15 | 0.27 | 0.71 | 0.35 | 0.11 | 0.15 | 0.29 |
| GPT-3.5 | 0.01 | 0.24 | 0.79 | 0.20 | 0.44 | 0.64 | 0.39 |
| Llama3-8B-Instruct | 0.66 | 0.48 | 0.80 | 0.71 | 0.30 | 0.73 | 0.61 |
| Llama2-7B-Chat Touvron et al. (2023) | 0.00 | 0.16 | 0.54 | 0.66 | 0.88 | 0.61 | 0.48 |
| Mistral-7B-Instruct-v0.2 Jiang et al. (2023a) | 0.28 | 0.06 | 0.92 | 0.40 | 0.26 | 0.16 | 0.35 |
| Gemma-1.1-7B-Instruct Team et al. (2024) | 0.31 | 0.68 | 0.03 | 0.35 | 0.67 | 0.68 | 0.45 |
| Qwen1.5-7B-Chat Bai et al. (2023) | 0.08 | 0.24 | 0.20 | 0.34 | 0.36 | 0.58 | 0.30 |

ask the model to generate a paragraph with the same number of prepositions as the given text. We summarize the results in Table 2 and the results are still not very satisfactory, showing the weaknesses of these LLMs.

## 4.4 REUSE LLM'S GENERATED CONTENT TO PERFORM OTHER TASK

In this section, we will discuss how to use LLM's previously generated content to evaluate new tasks.

### 4.4.1 GRAMMAR

Testing models on their understanding of word parts of speech within sentences is crucial because it reflects their grasp of grammar. Part of speech (POS) tagging (Gimpel et al., 2011) involves identifying whether a word functions as a noun, verb, preposition, etc., based on its usage in context. This understanding is fundamental to processing and generating coherent language, as it affects how words are combined to form meaningful sentences. A model's ability to accurately perform POS tagging indicates its proficiency in syntactic analysis, which is essential for any language-related task. The model is first asked about the number of prepositions in its previously generated content. Then we cut the first sentence in the previous paragraph and paste it at the last and generate a new paragraph, this operation preserves the number of prepositions. Then we ask the model in a separate run about the number of prepositions in the newly generated paragraph. We test on 100 samples and the initial paragraph is taken from the total word counting task in section 4.2.1. A schematic view can be found in Figure 2.

### 4.4.2 BASIC SQL TYPE OPERATIONS

Testing a model's ability to perform basic SQL operations based on input sentences not only evaluates its capacity to understand and manipulate data but also sheds light on its grasp of the finer structural details of sentences. This type of assessment requires the model to parse complex sentence structures and understand their relational dynamics to accurately convert natural language instructions into SQL commands. Successfully managing this translation indicates a deep understanding of syntax and semantics, reflecting the model's sophistication in language processing. Thus, proficiency in this area demonstrates more than just technical capability; it highlights the model's comprehensive linguistic competence, essential for any application involving natural language understanding and interaction. We use the generated texts in section 4.2.2. For each paragraph, we first ask the model to answer what is its $i$-th word, where $i$ is a randomly selected small integer. We then design the following tasks:

- Add first word: Add a random word to the beginning of the paragraph and ask the model what its $i + 1$-th word is. Then check whether the word is consistent with the previously answered one.

- Delete first word: Delete the first word of the paragraph and ask the model what its $i - 1$-th word is. Then check whether the word is consistent with the previously answered one.

- Change: Change the $i$-th word of the paragraph to $x$ and ask the model what its $i$-th word is. Then check whether the answer is $x$.

From the results in Table 3, we can see that all models have at least one task that performs badly, showing that they lag behind humans in these simple but fundamental tasks.

Table 3: Self-knowledge score using existing content.

| Model | Grammar | Add first word | Delete first word | Change | Avg |
|---|---|---|---|---|---|
| GPT-4 OpenAI (2023) | 0.30 | 0.63 | 0.59 | 0.40 | 0.48 |
| GPT-3.5 | 0.08 | 0.17 | 0.15 | 0.25 | 0.16 |
| Llama3-8B-Instruct | 0.62 | 0.51 | 0.68 | 0.20 | 0.50 |
| Llama2-7B-Chat Touvron et al. (2023) | 0.93 | 0.99 | 0.94 | 0.00 | 0.72 |
| Mistral-7B-Instruct-v0.2 Jiang et al. (2023a) | 0.20 | 0.42 | 0.53 | 0.08 | 0.31 |
| Gemma-1.1-7B-Instruct Team et al. (2024) | 0.45 | 0.55 | 0.67 | 0.04 | 0.43 |
| Qwen1.5-7B-Chat Bai et al. (2023) | 0.31 | 0.34 | 0.48 | 0.16 | 0.32 |

Table 4: Self-knowledge scores on multimodal tasks.

| Model | Counting | Color | Position | Avg |
|---|---|---|---|---|
| Gill Koh et al. (2023) | 0.06 | 0.45 | 0.46 | 0.32 |
| SEED-LLaMa Ge et al. (2023) | 0.26 | 0.81 | 0.53 | 0.53 |

## 5 EVALUATING THE SELF-KNOWLEDGE OF LMMS

### 5.1 IMPLEMENTATION DETAILS

There are only a few large multimodal models (LMMs) that can both understand and generate images when given textual instructions. Therefore, we just utilize two well-known LMMs that are trained to align vision encoder (e.g., ViT Dosovitskiy et al. (2020)), LLM, and vision decoder (e.g., diffusion model Ho et al. (2020)): Gill Koh et al. (2023) and SEED-LLaMa Ge et al. (2023). We also follow their default generation strategy in our tasks.

### 5.2 EXPERIMENTS

Perception is one of the most fundamental capabilities of LMMs, and the lack of perception will easily lead to the object hallucination problem Fu et al. (2023). Therefore, we consider several coarse-grained and important perception tasks for the self-knowledge evaluation of LMMs, including counting, color, and position. In particular, counting measures the LMMs' ability to determine the number of objects, color assesses how LMMs perceive specific colors, and position evaluates how LMMs recognize objects' spatial location and arrangement. For our experiments, we first prompt the LMMs to generate specific images, and then use the generated images for further evaluation. The instructions are shown in Table 9 in the Appendix.

Our experimental results reveal that SEED-LLaMa Ge et al. (2023) exceeds Gill Koh et al. (2023) on these self-knowledge tasks. SEED-LLaMa also demonstrates satisfactory performance in color generation and perception with a high score of 0.81. Besides, we notice that both the two LMMs gain poor performance on the counting task.

## 6 MORE DISCUSSIONS

### 6.1 ANALYZE THE BEHAVIOR OF SELF-KNOWLEDGE DESIGNATED WORD COUNTING TASK

Analyzing the underlying reason why LLM performs poorly on self-knowledge tasks is difficult. We make an attempt to analyze the case of the "designated word counting task", which has some special structure. Recall that this task requires the model to generate a keyword $x$ exactly $s$ times. When a human is asked to perform this task, whenever they generate a new word they will may some of their focus on whether this word is $x$ and how many times $x$ has appeared. In the context of LLM, we will use attention score to measure the extent of "focus". For each token in the generated paragraph, we extract the attention score of token $x$. We then sort these scores and only keep the top $15\%$ tokens as a set $\tau_x$. Then denote the number of times $x$ appears in the generated paragraph as $k = |x \in \tau_x|$. Then it is natural to define the attention-based score as $\frac{\min\{k,s\}}{\max\{k,s\}}$. To alleviate the influence of different attention heads, we average the attention score of the last layer's attention heads, we summarize

Table 5: Scores of designated word counting tasks.

| Model | Qwen1.5-7B | Mistral-7B | Gemma-1.1-7B | Llama2-7B | Llama3-8B |
|-------|-----------|-----------|-------------|-----------|-----------|
| Initial | 0.10 | 0.13 | 0.24 | 0.34 | 0.39 |
| Attention-based | 0.31 | 0.32 | 0.16 | 0.38 | 0.35 |
| Difference | 0.21 | 0.19 | 0.08 | 0.04 | 0.04 |

Table 6: Self-knowledge score under different evaluation protocols on the total word counting task.

| Model | No context eval | In-context eval | In-context eval with noise |
|-------|----------------|-----------------|---------------------------|
| GPT-4 OpenAI (2023) | 0.03 | 1.00 | 0.95 |
| GPT-3.5 | 0.00 | 0.90 | 0.96 |
| Llama3-8B-Instruct | 0.00 | 0.00 | 0.00 |
| Llama2-7B-Chat Touvron et al. (2023) | 0.00 | 0.00 | 0.00 |
| Mistral-7B-Instruct-v0.2 Jiang et al. (2023a) | 0.00 | 0.87 | 0.62 |
| Gemma-1.1-7B-Instruct Team et al. (2024) | 0.00 | 1.00 | 0.45 |
| Qwen1.5-7B-Chat Bai et al. (2023) | 0.01 | 0.70 | 0.89 |

the result in Table 5. We can see the difference between the initial self-knowledge score and the attention-based score is smaller when the initial self-knowledge score is bigger. This may imply that models that perform better at the initial self-knowledge task may behave more similarly to humans. But even the model that performs best still lags behind humans. This may be attributed to a human's strong ability to concentrate when asked to perform this task. There may be an *additive effect*: When the model's self-knowledge score is very poor, the poor performance may be mainly due to misalignment with this attention-based *mechanism*. When the self-knowledge score gets larger, it aligns with this attention-based mechanism, the poor performance may be attributed to the less-concentrates of LLM attention than humans. That is though the *mechanism* may be similar, the extent of attention score focusness is less than human.

## 6.2 DIFFERENT EVALUATION PROTOCOLS

The evaluation in the previous sections will make the generation process and evaluation process in separate runs. This evaluation process may become much easier when the generation process is given in the context, and we will call this evaluation protocol the in-context eval. As the in-context memory may make the evaluation too simple, recall that humans may starts to forget things when they are exposed to many irrelevant information. We consider the simplest setting where a short noise paragraph about 7000 tokens long is inserted between the generation process and the evaluation process. We call this in-context eval with noise. We summarize the result in Table 6. To our surprise, only GPT-4 and Gemma achieve 100% accuracy in the in-context eval and their performances are reduced when exposed to noise, similar to humans. Note some models like GPT-3.5 and Qwen may even have increased performance when exposed to noise. We conjecture that this weird phenomenon may be attributed to that some weak association is amplified due to stochastic resonance (Moss et al., 2004). But the main point here is that adding noise can reduce the performance of a *perfect* in-context evaluator, similar to the behavior of humans.

## 6.3 FINE-TUNING ON THE GENERATED DATA

We are also interested in the following question: What will happen if the model fine-tunes on its own generated contents? We mainly focus on the mathematics-related aspect as there has a standard benchmark GSM-8k (Cobbe et al., 2021) and it reflects the reasoning and language understanding of LLMs.

We conduct supervised fine-tuning for open-sourced LLMs, including Llama3-8B-Instruct, Llama2-7B-Chat Touvron et al. (2023), and Gemma-1.1-7B-Instruct Team et al. (2024). We train LoRA adapters Hu et al. (2021) for efficient fine-tuning. We utilize 4 24GB-4090 GPUs for three epoch training. The AdamW optimizer is used with a 1e-4 learning rate and the LoRA parameters dimension, alpha, and dropout are set to 64, 16, and 0.1, with a batch size of 16. For close-sourced LLM, we also use OpenAI API to fine-tune GPT-3.5 (gpt-3.5-turbo-1106) as GPT-4 is not yet available for

Table 7: GSM-8k accuracies.

| Model | Initial | Llama3 correct | Llama3 wrong | GPT-3.5 correct | GPT-3.5 wrong | Llama2 correct | Llama2 wrong |
|-------|---------|----------------|--------------|-----------------|---------------|----------------|--------------|
| Llama3 | 76.72 | 79.80 | 78.58 | 78.47 | 77.90 | 78.13 | 77.41 |
| GPT-3.5 | 71.38 | 71.08 | 70.62 | 71.42 | 71.23 | 71.23 | 71.23 |
| Llama2 | 24.11 | 24.41 | 24.03 | 25.32 | 24.26 | 24.91 | 25.32 |

finetuning for the public. We set the epoch to 3, with a batch size of 16 and a learning rate multiplier of 0.03.

We first consider two types of data, one is the "wrong one" directly generated by LLMs that is not human-checked and another is the correct one that has its answer human-corrected. We fine-tune each LLM on the data generated by itself and evaluate it on GSM-8k to get results in Figure 3. The initial accuracy on GSM-8k is: GPT-3.5: 71.38; Llama3: 76.72; Gemma: 48.07; Llama2: 24.11. We find models with higher initial accuracy will have higher accuracy when tuned on the correct answer and vice versa when the accuracy is low. This is similar to humans as people may not distinguish good and bad when they are not good at something, but when their ability increases, they start to have their own judgments. Note all models have improved accuracies when tuning on its own data except GPT-3.5 when tuning on the wrong data. As GPT-3.5's black-box tuning nature, we attribute this as an outlier.

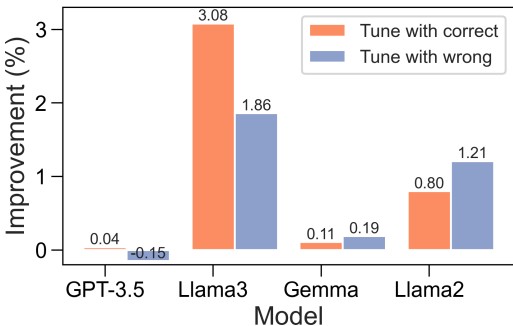

Figure 3: GSM-8k accuracy after fine-tuning on different data.

To further see the influence of tuning on other's generated data. We consider GPT-3.5; Llama3 and Llama2 that have similar architecture. We consider tuning on both the correct and wrong data and summarize the results in Table 7. We find that model achieves its highest accuracy when tuning on its self-generated content and the content generated by models that have higher accuracy may not guarantee the highest improvements. this may suggest that self-improving is a promising direction to further enhance model capacity.

## 7 CONCLUSION

In this paper, we introduce a comprehensive self-knowledge evaluation framework specifically designed for large language models (LLMs) and large multimodal models (LMMs). This framework focuses on assessing their capacity to comprehend and effectively respond to questions they generate themselves. Our extensive findings across a variety of tasks demonstrate that these models still exhibit significant deficiencies when engaged in self-knowledge tasks. Further analysis suggests that these shortcomings may be partially attributed to a misalignment with human attention mechanisms and thus highlight a potential area for improvement. Additionally, we explore the impact of fine-tuning models on self-generated data, which shows promise in enhancing their overall performance. Our framework provides a valuable and efficient approach to advancing both the evaluation and development of LLMs and LMMs. While we primarily consider cases that are straightforward and easily verified by humans, future work could involve increasing the complexity and automation of the evaluation challenges.

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

Table 8: The list of verifiable instructions, with brief descriptions. We use these instructions because we think they are either easy to verify or common in real-world applications.

| Task | First Generate | Then Evaluate |
|---|---|---|
| Total count | Generate a paragraph with exactly [num] words in total. | How many words are there in the following paragraph? paragraph |
| Designate count | Generate a paragraph where the [word] appears exactly [num] times. | How many times does the [word] appear in the following paragraph? paragraph |
| Facts | Name a celebrity that was born on [year, month, day]. | Is the following statement true? paragraph |
| ArXiv | Give me a paper with its title and arXiv ID, which was submitted on [year, month]. | What is the arXiv ID of the paper titled [title]? |
| Math | Generate a hard high school level mathematics question with [answer]. | Is [answer] the correct answer to the following question? question |
| Theorem | Generate a hard elementary one variable inequality proving problems rigorously and clearly, no need to generate the proof. | Is the following inequality true? inequality |
| Code | Generate a hard coding problem in Python. The code's execution result should be [answer]. | What is the execution result of the following code? code |

Table 9: The list of verifiable instructions for LMMs.

| Instruction Group | First Generate | Then Evaluate |
|---|---|---|
| Counting | Generate an image with exactly [num] [objects]. | How many [objects] are there in the image? image |
| Color | Generate an image with a [color] [object]. | What's the color of [object] in the image? image |
| Position | Generate an image with a computer [position relationship] a [object]. | Is the computer [position relationship] a [object] in the image? image |

# Appendix

## A  PROMPTS

We summarize some of the used prompts in this Appendix. Please refer to Table 8 and 9.

## B  MORE EXPERIMENTS

### B.1  AGENT

The tasks used in Table 8 are handcrafted by humans, so it remains interesting to see that AI agents generate questions in this manner. If this is possible, it will make AI autonomously generate questions beyond human-designed ones and may pave ways to self-verify and self-improvement without human supervision.

We use two GPT-4 as agents, one as a question generator, and another as a judge. To let the agent understand our goal, we feed the handcrafted data in Table 8 to the agent and ask it to generate tasks in this manner. The judge is asked to decide whether the question generated by the previous agent is clear and has a unique answer that can be easily verified. Interestingly, we can get some template questions, we summarize some in Table 10. One can further ask the model to generate for example

Table 10: Selected template questions generated by AI agents.

| Task | First Generate | Then Evaluate |
|---|---|---|
| Specific Mention | Write a paragraph mentioning exactly [num] distinct [countries]. | Are there exactly [num] distinct [countries] mentioned in the following paragraph? paragraph |
| Sentiment Analysis | Write a paragraph where the overall sentiment is positive, with exactly [num] positive words. | Is the overall sentiment of the following paragraph positive with exactly [num] positive words? paragraph |
| Sports Statistics | Provide statistics for a [sport] game played on [date]. | Are the following statistics correct for the [sport] game played on [date]? statistics |

Table 11: Scores on MMLU college cs tasks.

| Model | Qwen1.5-7B | Mistral-7B | Gemma-1.1-7B | Llama2-7B | Llama3-8B | GPT-3.5 | GPT-4 |
|---|---|---|---|---|---|---|---|
| Initial | 0.42 | 0.51 | 0.47 | 0.40 | 0.50 | 0.59 | 0.81 |
| Self-knowledge | 0.46 | 0.45 | 0.59 | 0.54 | 0.42 | 0.52 | 0.75 |

100 instances of questions based on the template question. This shows that agents have the potential to work without human supervision, we leave the detailed investigation in this direction as future work.

### B.2 EXISTING BENCHMARK BASED SELF-KNOWLEDGE

As our self-knowledge evaluations in previous sections are mostly based on our manually created template problems, one may wonder if we can leverage the existing human-crafted benchmarks to perform self-knowledge evaluations. Of course, one may also use the dual-generating framework in section 4.3. In this section, we introduce another way which may be more efficient. Wang et al. (2022) introduce the philosophy of augmenting the instruction tuning data using LLMs. Motivated by this philosophy, we consider showing the LLM the test data from a benchmark and letting it generate new testing problems with answers and we then let the LLM do these self-generated problems. We consider the widely adopted benchmark MMLU (Hendrycks et al., 2020) and as it consists of too many topics, we choose the college cs task for simplicity. We summarize the results in Table 11. We find that the difference between the initial accuracy and self-knowledge score is small. More interestingly, it seems that when a model has its initial accuracy greater than $50\%$ it will have its initial accuracy greater than the self-knowledge accuracy and vice versa. This is similar to Figure 3 where models with higher accuracy may favor the correct answered data.

## C ABLATION STUDIES

### C.1 ABLATION STUDY ON INCONSISTENCY

Recall that equation (3) depicts a way to assess the self-knowledge ability through consistency. Similarly, if a transformation $\hat{\tau}$ will always make the answer to task $T$ **changed** when applying $\hat{\tau}$ to $\mathbf{x}$, then the self-knowledge score can be calculated via the **inconsistency**.

$$\mathbb{I}(\text{LLM}(\text{question-verifying prompt}, \mathbf{x}) \neq \text{LLM}(\text{question-verifying prompt}, \hat{\tau}(\mathbf{x}))). \qquad (6)$$

We consider the Math and Fact tasks, where the operation $\hat{\tau}$ is easy to construct. For example, reduce the generated date by one day. From the results in Table 12, we can see that all models cannot perform well on both the consistency-based and inconsistency-based self-knowledge checks. This further supports our conclusion that the model does not really understand its generated content.

Table 12: Self-knowledge score under consistency and inconsistency.

| Model | Fact (Consistency) | Fact (Inconsistency) | Math (Consistency) | Math (Inconsistency) |
|---|---|---|---|---|
| GPT-4 OpenAI (2023) | 0.71 | 0.35 | 0.24 | 0.99 |
| GPT-3.5 | 0.68 | 0.25 | 0.58 | 0.72 |
| Llama3-8B-Instruct | 0.30 | 0.70 | 0.14 | 0.99 |
| Llama2-7B-Chat Touvron et al. (2023) | 0.65 | 0.43 | 0.88 | 0.52 |
| Mistral-7B-Instruct-v0.2 Jiang et al. (2023a) | 0.92 | 0.04 | 0.23 | 0.88 |
| Gemma-1.1-7B-Instruct Team et al. (2024) | 0.15 | 0.84 | 0.93 | 0.75 |
| Qwen1.5-7B-Chat Bai et al. (2023) | 0.77 | 0.26 | 0.57 | 0.97 |

Table 13: Detailed accuracies on generation and verification.

| Model | Initial | Gen | Verify | True | Initial (Expert) | Gen (Expert) | Verify (Expert) | True (Expert) |
|---|---|---|---|---|---|---|---|---|
| GPT-4 | 0.03 | 0.08 | 0.00 | 0.00 | 0.19 | 0.01 | 0.00 | 0.00 |
| GPT-3.5 | 0.00 | 0.06 | 0.00 | 0.00 | 0.05 | 0.05 | 0.01 | 0.00 |
| Llama3 | 0.00 | 0.02 | 0.00 | 0.00 | 0.03 | 0.06 | 0.01 | 0.00 |
| Llama2 | 0.00 | 0.00 | 0.00 | 0.00 | 0.00 | 0.01 | 0.00 | 0.00 |
| Mistral | 0.00 | 0.01 | 0.00 | 0.00 | 0.63 | 0.00 | 0.00 | 0.00 |
| Gemma | 0.00 | 0.02 | 0.00 | 0.00 | 0.02 | 0.00 | 0.00 | 0.00 |
| Qwen | 0.01 | 0.01 | 0.00 | 0.00 | 0.12 | 0.00 | 0.01 | 0.00 |

## C.2 ABLATION STUDY ON PROMPT

### C.2.1 EXPERT PROMPT

To evaluate the influence of role-modeling prompts on the experiments, we conduct similar experiments to those in section 4.2.1 by changing the question-generating prompt to "Assume you are an expert in counting numbers. Generate a paragraph with exactly [num] words in total." and the question-verifying prompt to "Assume you are an expert in counting numbers. How many words are there in the following paragraph?". In Figure 4, we can see adding the expert prompt indeed improves the self-knowledge score showing that expert role modeling has some positive influence. Note the drastic improvement of Mistral is due to similar reasons of in-context eval in Table 6. The model encodes a "cheat sheet" like "This paragraph, my friends, consists of precisely 58 words." to help the self-verifying process.

To investigate further the impact of the prompt, we will the true answer into account. Specifically, we calculate the ground-truth number of words in the generated paragraph and calculate the following three accuracies: Gen: The accuracy that the generated content has the required number of words. Ver: The accuracy that the verify answer from the model on the generated content is equal to the real number of words in the generated content. True: The accuracy that the verify answer from the model on the generated content is equal to the real number of words in the generated content and also equal to the required number of words when generating it. We summarize the results in Table 13. We found that none of the real generative accuracy or verifying accuracy is improved when using the expert prompt, showing a deep underlying reason behind the improvements in self-knowledge score, we leave the investigation of the underlying reasons as future work.

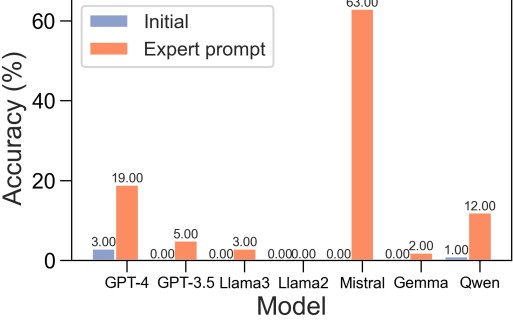

Figure 4: The effect of expert prompt.

Table 14: Self-knowledge score with and without CoT.

| Model | Code (w/o CoT) | Code (w CoT) | Math (w/o CoT) | Math (w CoT) |
|---|---|---|---|---|
| GPT-4 OpenAI (2023) | 0.08 | 0.11 | 0.24 | 0.15 |
| GPT-3.5 | 0.51 | 0.46 | 0.58 | 0.35 |
| Llama3-8B-Instruct | 0.68 | 0.79 | 0.14 | 0.02 |
| Llama2-7B-Chat Touvron et al. (2023) | 0.16 | 0.14 | 0.88 | 0.94 |
| Mistral-7B-Instruct-v0.2 Jiang et al. (2023a) | 0.07 | 0.11 | 0.23 | 0.15 |
| Gemma-1.1-7B-Instruct Team et al. (2024) | 0.42 | 0.43 | 0.93 | 0.85 |
| Qwen1.5-7B-Chat Bai et al. (2023) | 0.84 | 0.90 | 0.57 | 0.34 |

### C.2.2    CHAIN-OF-THOUGHT PROMPTING

We also test the influence of another popular prompting strategy chain-of-thought (CoT) prompting Wei et al. (2022). We use CoT in both generative and verify processes just as Section C.2.1 and we summarize the results in Table 14. We find that CoT does not always improve the self-knowledge score unlike the expert prompt.

