# OpenReview forum: "Can I Understand What I Create? Self-Knowledge Evaluation of Large Language Models"
_ICLR.cc/2025/Conference — Submitted to ICLR 2025_

### Official Review · Reviewer_m4Xw · 2024-10-19

**Soundness:** 2
**Presentation:** 2
**Contribution:** 2
**Rating:** 3
**Confidence:** 4

**Summary:**

The paper discusses the topic: "Can Large Language Models (LLMs) Understand What They Create?" Essentially, in the paper, LLMs are asked to generate questions on a given topic and provide answers to those questions. If the model can answer all the self-generated questions correctly, it is said to have self-knowledge evaluation ability. The paper also aims to provide insight into the model's self-knowledge capabilities. Additionally, it discusses the self-knowledge evaluation framework of Large Multimodal Models (LMMs).

**Strengths:**

1. The paper introduces an interesting research question: "Can Large Language Models (LLMs) be evaluated by the questions they generate themselves?"
2. Experimental results show that existing LLMs and LMMs do not perform satisfactorily in self-knowledge evaluation. Only GPT-4 and Gemma-1.1-7B-Instr demonstrated perfect self-knowledge evaluation ability when the question generation process was given in context.
3. The authors also attempt to provide insights into the limitations of the models' self-knowledge abilities.

**Weaknesses:**

1. The authors evaluated the models' self-knowledge abilities across seven different tasks, but it is unclear why these specific tasks were chosen. Does this set of tasks cover all aspects of self-knowledge evaluation, or does this selection follow some other criteria and existing literature (e.g., MMLU)?
2. The authors provide insight into the limitations of the models' self-knowledge ability (lines 421–459), which is good. However, the insights are based solely on **a single task (the word-counting task) out of the seven**, which is insufficient for me to provide a general understanding of the limitations of the models' self-knowledge abilities.
3. In my opinion, the paper is not well-organized. For instance, Section 5, Evaluation of LMMs, is not well-connected with the preceding or subsequent sections. The content in Section 5 feels somewhat isolated from the rest of the paper. And the experimental details in Section 5 are not explained clearly. A better flow and connection between sections would improve the readability and understanding of the paper.

**Questions:**

1. The authors claim that GPT-4 and Gemma-1.1-7B-Instr have shown perfect self-knowledge evaluation ability when the question generation process is given in context (lines 60–63). However, this seems odd to me, as GPT-4 and Gemma-1.1-7B-Instr differ significantly in terms of parameter capacity, training data, architecture, etc. Can you provide some insights into why these two models demonstrated perfect self-knowledge evaluation ability? Why didn't other models, like LLaMA-3.1-8B, which has a similar parameter capacity to Gemma-1.1-7B-Instr, show perfect self-knowledge evaluation ability?

2. I am curious about the motivation behind the research question: "Can Large Language Models (LLMs) be evaluated by the questions they generate themselves?" In my understanding, there are two possible outcomes: a) the model can perfectly answer the self-generated questions, or b) the model can partly answer the self-generated questions. In the first case, if the model can perfectly answer the self-generated questions, in my opinion, this framework may not help us identify the model's limitations, which is the main goal of evaluation of LLMs. In the second case, it becomes difficult to determine whether the model failed because of its abilities or because the generated questions don't make sense. Can you provide some insights on this?

---

### Official Review · Reviewer_utKS · 2024-11-01

**Soundness:** 2
**Presentation:** 1
**Contribution:** 2
**Rating:** 3
**Confidence:** 3

**Summary:**

This paper proposes to evaluate LMs with data they themselves generate (e.g. consistency). For example, given a prompt to generate something (a paragraph with 56 words) they then check if the model thinks that that generation fulfills this task. They have the LM generate and verify it's generations on 6 categories: counting words (2x categories), facts, grammar, math, and code.

They show that models do poorly in these categories, and show some interesting results when random noise is added between the generation and discriminative sections. Finally, they show that fine-tuning on the math task improves the math performance.

**Strengths:**

- The paper's proposal to self-evaluate makes it easier to evaluate LMs with needing to have a gold answer set.
- The paper evaluates many models, including many SoTA ones.

**Weaknesses:**

1. My biggest concern is I am not sure what novelty this paper contributes. They summarize it as "We find that modern LLMs and LMMs have unsatisfactory behaviors on self-knowledge evaluations" which I agree is their main contribution but also one that has been widely observed in the community [1-3].
2. The paper focuses a lot on the word counting task, but this also has been proposed and studied before [4-5].
3. Nearly every result in the paper has been studied somewhere: RAG-based papers have looked into the noisy context problem [6-7], while fine-tuning on current generations is the basis for lots of RL-based tuning.
4. There are lots of grammar issues in the paper, such as `words in total.”.`  (two periods) on line 141, "The self-generate step" on line 99, "We can also make further verify" on line 312, and many more, as well as citations being inline rather than in parentheses (minor).

I don't disagree with any of the results in the paper, but the paper is in a somewhat rough shape presentation-wise and doesn't seem to present any new findings.

[1] The Generative AI Paradox: "What It Can Create, It May Not Understand"

[2] SELF-[IN]CORRECT: LLMs Struggle with Discriminating Self Generated Responses

[3] Beyond Probabilities: Unveiling the Misalignment in Evaluating Large Language Models

[4] Following Length Constraints in Instructions

[5] Large Language Models Lack Understanding of Character Composition of Words

[6] Making Retrieval-Augmented Language Models Robust to Irrelevant Context

[7] Entity-Based Knowledge Conflicts in Question Answering

**Questions:**

The main question in the Weaknesses: what does your work contribute that previous work has not?

---

### Official Review · Reviewer_Put9 · 2024-11-01

**Soundness:** 1
**Presentation:** 1
**Contribution:** 1
**Rating:** 1
**Confidence:** 5

**Summary:**

This paper introduces a benchmark centered around the "self-knowledge" of LLMs, prompting them to generate some content that should have a given task label, then in a separate prompt having them perform the task on the generated content. The benchmark scores LLMs for how frequently the prediction from the second prompt matches the original task label.

The authors run a series of tests centered around what they call this "first generate .. then evaluate" protocol. They find that the consistency score is relatively low for many current LLMs.

**Strengths:**

* The authors consider a series of task scenarios and experiment with many LLMs, both open and closed source.
* The paper is somewhat well-motivated, as we want our LLMs to be able to reliably answer the problems that they synthesize in a given domain.

**Weaknesses:**

**W1** The paper is very badly written. It is littered with malformed sentences, bad English and unclear writing. I generally give this a pass when I review conference submissions, but the extent of the typos and grammar errors made it very hard to read.

**W2**  The evaluation protocol has far too many degrees of freedom. It is unclear for each subtask whether (A) the answer the LLM generates in the initial paragraph is incorrect, (B) whether its prediction for the second half is incorrect, (C) both correct or (D) both incorrect. While the metric "does the LLM give the same answer both times?" gives a partial signal for whether the LLM is making a mistake (since if they don't match, one prediction surely is wrong), it could be the case that both are wrong. The evaluation would mark the (D) scenario as correct since it is consistent even though it is wrong.
   * The "dual-generating" evaluation in 4.3 adds another source of noise: if the translation between $x$ and $x'$ is not faithful, then it obscures whether the agreement between $a$ and $a'$ actually reflects the model's ability to perform the task in question.
   * Another confounding factor: different LLMs will generate different complexities of $x$'s for each task. Simple, unconstrained prompts like "Generate a paragraph" or "generate a hard coding problem" will get different kinds of outputs from different LLMs. If GPT-4 generates programs that are way harder than the programs generated by Gemma, then it is impossible to compare the resulting consistency metric for GPT-4 to that of Gemma. The authors should consider how to reformulate the evaluation so that it can more reliably compare results between models.

**W3**: Section 5 (which is mis-labeled) is poorly explained and not self-contained, as it relies completely on the content in the appendix.

**W4**: Section 6.3 is not related to the rest of the paper. It uses a completely different dataset (GSM) and has little to do with the author's discussed notion of self evaluation.

**Questions:**

**Q1** Did you explore consistency in the results with respect to different prompting methods? E.g. different ways to word the task instructions, or using $n$-shot prompting instead of 0-shot? The only prompt variability I see is the section where you (somewhat inexplicably) added 7000 tokens of noise.

**Q2** Did you verify the statistical significance of the results?

**Q3** Please provide an explanation for why all models failed on the "total count" task-- it is hard to believe that they all got 0-3% consistency. Releasing/attaching your experimental code would also make this easier to believe.

---

### Official Review · Reviewer_EaMN · 2024-11-03

**Soundness:** 2
**Presentation:** 2
**Contribution:** 2
**Rating:** 3
**Confidence:** 4

**Summary:**

This paper presents a framework for evaluating large language models (LLMs) based on their self-knowledge, aiming to assess model's evaluation capability over its own generated solutions. They reveal there is an unsatisfactory performance on self-knowledge evaluation.

**Strengths:**

1. Provide a framework to perform self-knowledge evaluation.

**Weaknesses:**

1. The paper does not fully explain why self-knowledge is critical for LLMs or how it relates to practical AI applications. Your paper showed LLMs or LMMs exhibit deficiencies when evaluating some designed self-knowledge task. But what is the impact of this deficiencies to LLM application? I am still confused about motivation of this project.

2. Most of evaluation tasks are toy tasks (such as word counting) and are exited to evaluate LLM. This paper simply built a special case of self knowledge evaluation

3. Although multimodal models are tested, the tasks are limited to basic perception capabilities, like counting and color identification.

**Questions:**

See weakness

Can you demonstrate the difference of self-knowledge evaluation and evaluation of other model generated texts? Currently, I cannot distinguish LLM simply cannot self-evaluate or evaluate those tasks in general.

There are prior works show LLMs have self-bias to its own generated texts. I think they are relevant.
https://arxiv.org/abs/2404.13076
https://aclanthology.org/2024.acl-long.826.pdf
https://arxiv.org/pdf/2311.09766

---

### Meta-Review · Area_Chair_ezBk · 2024-12-20

**Metareview:**

The paper investigates the evaluation of self-knowledge.  This is an important area of research trying to probe how much an LLM is aware of its self knowledge.   Reviewers point out several weakness:  poor structure of the paper, less novelty with respect to earlier work, limited evaluation settings and so fort.  Given the consensus between reviewers, I suggest that the paper be rejected.

**Additional Comments On Reviewer Discussion:**

There was no discussion between reviewers and authors.

---

### Decision · Program_Chairs · 2025-01-22

Reject